# hiPSC-Derived Schwann Cells Influence Myogenic Differentiation in Neuromuscular Cocultures

**DOI:** 10.3390/cells10123292

**Published:** 2021-11-24

**Authors:** Sarah Janice Hörner, Nathalie Couturier, Roman Bruch, Philipp Koch, Mathias Hafner, Rüdiger Rudolf

**Affiliations:** 1Institute of Molecular and Cell Biology, Mannheim University of Applied Sciences, 68163 Mannheim, Germany; s.hoerner@hs-mannheim.de (S.J.H.); n.couturier@hs-mannheim.de (N.C.); r.bruch@hs-mannheim.de (R.B.); m.hafner@hs-mannheim.de (M.H.); 2Interdisciplinary Center for Neurosciences, Heidelberg University, 69120 Heidelberg, Germany; 3Central Institute of Mental Health, Medical Faculty Mannheim of Heidelberg University, 68159 Mannheim, Germany; philipp.koch@zi-mannheim.de; 4Hector Institute for Translational Brain Research (HITBR gGmbH), 68159 Mannheim, Germany; 5German Cancer Research Center (DKFZ), 69120 Heidelberg, Germany; 6Institute of Medical Technology, Mannheim University of Applied Sciences and Heidelberg University, 68163 Mannheim, Germany

**Keywords:** AChR, acetylcholine receptors, hiPSC, in vitro, neural crest, neuromuscular junction, NMJ, Schwann cells, stem cells

## Abstract

Motoneurons, skeletal muscle fibers, and Schwann cells form synapses, termed neuromuscular junctions (NMJs). These control voluntary body movement and are affected in numerous neuromuscular diseases. Therefore, a variety of NMJ in vitro models have been explored to enable mechanistic and pharmacological studies. So far, selective integration of Schwann cells in these models has been hampered, due to technical limitations. Here we present robust protocols for derivation of Schwann cells from human induced pluripotent stem cells (hiPSC) and their coculture with hiPSC-derived motoneurons and C2C12 muscle cells. Upon differentiation with tuned BMP signaling, Schwann cells expressed marker proteins, S100b, Gap43, vimentin, and myelin protein zero. Furthermore, they displayed typical spindle-shaped morphologies with long processes, which often aligned with motoneuron axons. Inclusion of Schwann cells in coculture experiments with hiPSC-derived motoneurons and C2C12 myoblasts enhanced myotube growth and affected size and number of acetylcholine receptor plaques on myotubes. Altogether, these data argue for the availability of a consistent differentiation protocol for Schwann cells and their amenability for functional integration into neuromuscular in vitro models, fostering future studies of neuromuscular mechanisms and disease.

## 1. Introduction

Neuromuscular junctions (NMJs) are synapses in the peripheral nervous system, where motor signals are transmitted from presynaptic nerve terminals to muscle fibers to induce muscle contraction and, therefore, control body movement. In humans as well as rodents, the core NMJ is a tripartite structure, built of cholinergic lower motoneurons (MN) extending from the spinal cord, nicotinic acetylcholine receptors in the sarcolemma, and specialized Schwann cells (SC), which cap the synapse (Figure 1). To date, a wealth of in vitro NMJ models and studies is available, and recent developments in the areas of microfluidics, 3D cell culture, and human induced pluripotent stem cell (hiPSC) technology have greatly increased their relevance for in vitro assays. Recently, these developments have been reviewed from different angles: Barbeau and colleagues provided a comprehensive overview of historical and recent developments of NMJ cocultures, ranging from first animal explant cultures to the latest developments with neuromuscular organoids [1]. Others focused on the potential of NMJ models for synaptogenesis and degeneration studies [2], disease modeling [3], and drug testing [4]. Lynch and colleagues reviewed in vitro models which include at least one cell component derived from human stem cells and emphasized the importance of human-based approaches [5]; Luttrell et al. reviewed state-of-the-art developments of culture formats integrating stem cell-derived in vitro NMJs [6]. Besides all promising developments that have improved the relevance of in vitro NMJ models, their major drawback was a lack of SC, and this was identified as a critical issue. Indeed, terminal SC have been shown to play crucial roles for NMJ development [7,8,9], maturation [10,11,12], and function [13,14] in vivo. In vitro, addition of a rat-derived SC line improved maturation in microfluidic neuromuscular cocultures [15]. The higher NMJ maturation levels which can be achieved in explant cultures [16] have been linked to the presence of glia co-differentiating with neurons in explants [17], and primary animal-derived spinal glia were added in neuromuscular cocultures for support of MN [18]. Studies presenting 3D organoids derived from hiPSC-derived neuromuscular cultures, where mixed cell populations were differentiated in parallel, showed the appearance of cells positive for the SC marker S100b [19]. Similar findings were reported from 2D cultures [20]. However, to date, no study has shown selective integration of hiPSC-derived SC into neuromuscular cocultures. This would allow cell type-specific studies and manipulations. Likely a major reason for this flaw is that only few differentiation protocols for stem cell-derived SCs have been available. In addition, most of them are rather time-consuming and offer only low cell yields, thus mostly necessitating cell sorting steps [21,22,23].

In this study, we present a protocol for setting up tricultures including hiPSC-derived SC combined with hiPSC-derived MN and C2C12 myotubes within nine days. To enable this, we further demonstrate derivation of SC from hiPSC with increased protocol robustness, based on a differentiation protocol presented by Kim and colleagues [24]. This paves the way for studying cell-autonomous processes and how the different cell types influence each other; it opens the possibility to study cell type-specific contributions in pathological scenarios by use of patient-derived hiPSCs, and it might be a further step towards creating more mature in vitro NMJs.

## 2. Materials and Methods

### 2.1. hiPSC Culture

hiPSC cultures were maintained in standard 6-well tissue culture plates coated with heSC-qualified Geltrex (Thermo Fisher Scientific, Darmstadt, Germany) in either E8 or mTeSR1 medium (STEMCELL Technologies, Cologne, Germany), with daily medium changes. hiPSC were split routinely with 0.5 mM EDTA (Thermo Fisher Scientific) upon reaching 80–90% confluency. Differentiation to motoneurons or Schwann cells was induced 2–3 days after routine EDTA splitting upon 50–60% confluency of hiPSC colonies by switching to differentiation medium, defining day 0 of differentiation. Throughout differentiation, cells were kept in Geltrex coated 6-well tissue culture plates, and 5 µM Y-27632 (STEMCELL Technologies) was added to the medium for no longer than 24 h after passaging or thawing. For hiPSC differentiation, cell lines 028#1 (provided by Philipp Koch, HITBR Hector Institute for Translational Brain Research, Mannheim) and WC035i-SOD1-D90D (purchased from WiCell Research Institute, Madison, WI, USA) were used. WC035i-SOD1-D90D was deposited with WiCell by Su-Chun Zhang, University of Wisconsin [25] and is derived from cell line ND29149 from the NINDS Human Genetics Resource Center at the Coriell Institute for Medical Research.

### 2.2. hiPSC Motoneuron (MN) Differentiation

The protocol for MN differentiation was based on Du et al., 2015 [26] and employs molecular factors CHIR99021, for Wnt activation; dual SMAD inhibitors SB431542 and DMH1, for neural induction; retinoic acid, for caudalization; Shh activator purmorphamine, for ventralization; and Notch inhibitor Compound E as well as neurotrophic factors BDNF, GDNF, and IGF-1, for neuronal maturation. hiPSC were switched to MN differentiation medium 1 (MNDM1), comprised of MN basal medium (BM) containing Neurobasal medium and DMEM/F12 medium without HEPES (Thermo Fisher Scientific) in a 1:1 ratio, 1% Penicillin-Streptomycin (Sigma Aldrich, Taufkirchen, Germany), 1% GlutaMAX (Thermo Fisher Scientific), 0.5 × NeuroCult SM1 Supplement (STEMCELL Technologies), and 0.5 × N2 Supplement (STEMCELL Technologies) and supplemented with 0.1 mM ascorbic acid (Carl Roth, Karlsruhe, Germany), 3 µM CHIR99021 (Tocris Bioscience, Bristol, UK), 2 µM DMH1 (Selleck Chemicals, Houston, TX, USA), and 2 µM SB431542 (Cell Guidance Systems, Cambridge, UK). MNDM1 was changed daily for 6 days. As cells became denser, medium volume per well was increased to prevent medium acidification between medium changes. On day 6, cells were passaged using Accutase (Thermo Fisher Scientific) and plated at a 1:6 ratio onto Geltrex coated 6-well plates in MNDM2, consisting of BM supplemented with 0.1 mM ascorbic acid, 1 µM CHIR99021, 2 µM DMH1, 2 µM SB431542, 0.5 µM purmorphamine (Cell Guidance Systems), and 0.1 µM all-trans retinoic acid (Sigma Aldrich). MNDM2 was changed daily until day 12, when cells were split into suspension culture using Dispase-II (Merck Millipore, Darmstadt, Germany) in 6-well suspension culture plates. Cells were split at a well ratio of 1:2 in 4–5 mL MNDM3 per well, consisting of BM supplemented with 0.1 mM ascorbic acid, 0.1 µM purmorphamine, 0.5 µM retinoic acid, 10 ng/mL BDNF, 10 ng/mL GDNF, and 10 ng/mL IGF-1 (Peprotech, Hamburg, Germany). Medium was changed every other day by carefully replenishing 75% of medium volume in each well. After 6 more days, cell aggregates were collected and dissociated into a single cell suspension with Accumax (Sigma Aldrich). Cells were seeded subsequently at a 1:1–1:3 ratio into PLL (Sigma Aldrich)/rhLaminin-521 (BioLamina, Sundbyberg, Sweden)-coated plates in MNDM4, consisting of BM supplemented with 0.1 mM ascorbic acid, 0.1 µM purmorphamine, 0.5 µM retinoic acid, 0.1 µM Compound E (Merck Millipore), 10 ng/mL BDNF, 10 ng/mL GDNF, and 10 ng/mL IGF-1. After each replating, ROCK1 inhibitor Y-27632 (Cell Guidance Systems) was added to the medium for 24 h at a concentration of 5 µM. For expansion and cryopreservation, motoneuron progenitors were either frozen on day 12 in BM containing 10% DMSO (Carl Roth) and 10 µM Y-27632 or expanded with Accutase at a 1:6 ratio every 6 days for 1–2 passages and frozen subsequently. For expansion and thawing, MNDM2 was additionally supplemented with 0.5 mM valproic acid (Sigma Aldrich), and after thawing, 10 µM Y-27632 was added for 24 h. Throughout the protocol, all media containing retinoic acid were kept away from direct light and stored in light protected containers, and direct light inside the biosafety cabinet was turned off when changing medium.

### 2.3. hiPSC Schwann Cell (SC) Differentiation

SC differentiation protocol was modified from Kim et al., 2017 [24] and employs molecular factors CHIR99021 and SB431542 for Wnt activation and neuralization; BMP activator BMP4 and BMP inhibitor DMH1 to precisely activate BMP signaling; and factors known to drive SC differentiation, which are neuregulinβ-1 (NRG1), adenylyl cyclase activator forskolin, retinoic acid, and PDGF-BB. First, hiPSC were differentiated to SC precursors in 24 days. Basal medium for SC precursor differentiation consisted of Neurobasal medium and Advanced DMEM/F12 (Thermo Fisher Scientific) in a 1:1 ratio, 1% Pen-Strep, 1% GlutaMAX, 1 × SM1 supplement, and 1 × N2 supplement. Differentiation was induced on day 0 by switching to SC precursor differentiation medium 1 (SCPM1), supplemented with 2 µM SB431542 and 3 µM CHIR99021. After 6 days, cells were split 1:10 with Accutase into SCPM2 supplemented with 2 µM SB431542, 3 µM CHIR99021, 50 ng/mL NRG1, 15 ng/mL BMP4, and 1 µM DMH1. On day 12, cells were split 1:6 into SCPM3 supplemented with 2 µM SB431542, 3 µM CHIR99021, and 50 ng/mL NRG1. Cells were maintained in SCPM3 until day 24, with another split on day 18. Then, base medium was switched to SC medium (SCM), consisting of DMEM/low glucose (Thermo Fisher Scientific) supplemented with 1% Pen-Strep, 1% GlutaMAX, 1% FBS, and 200 ng/mL NRG1. On day 24, SC precursors were split 1:3–1:6 in SCM additionally supplemented with 4 µM forskolin, 0.1 µM all-trans retinoic acid, and 10 ng/mL PDGF-BB. On day 28, SCM was supplemented with PDGF-BB but not with forskolin or retinoic acid anymore. From day 30 onwards, cells were maintained in SCM without PDGF-BB, forskolin, or retinoic acid. SC were allowed to grow to confluency and then passaged with Accutase at a 1:3 ratio. Keeping the cultures above 40% confluency increased cell survival. Cells were maintained up to day 100 and cryopreserved upon routine splitting. SC used for experiments were between days 50 and 90 of differentiation.

### 2.4. C2C12 Culture

C2C12 murine myoblasts were maintained in C2C12 growth medium (GM) consisting of DMEM/F12, high glucose, and L-glutamine, without sodium pyruvate (Sigma Aldrich) and supplemented with 20% FBS (Capricorn Scientific, Ebsdorfergrund, Germany). Cells were passaged every other day using trypsin (Sigma Aldrich) and were never allowed to grow beyond 60% confluency. Differentiation to myotubes was induced by switching confluent cultures to C2C12 differentiation medium (DM) supplemented with 2% horse serum (Thermo Fisher Scientific). Generally, only C2C12 below passage 20 were used. We noticed an influence of C2C12 passage specifically on acetylcholine receptor (AChR) cluster formation and, therefore, recommend using C2C12 of the same passage number for experiments which are to be compared. Data used for AChR cluster analysis were obtained from experiments using C2C12 at passage 12 at the timepoint of differentiation induction.

### 2.5. Tricultures

To obtain tricultures, cryopreserved C2C12 myoblasts, MN progenitors, and SC were thawed on the same day. C2C12 were routinely passaged and, two days prior to triculture seeding, plated in GM at a density of 70,000–80,000 cells per cm^2^ into Geltrex coated 6-well tissue culture plates with glass coverslips placed in the wells and switched to DM the following day. SC were cultured in SCM and passaged once with Accutase 4–5 days after thawing, depending on confluency. MN progenitors were grown for 3 days after thawing in MNDM2 containing valproic acid to obtain a dense cell layer and then split to suspension, as described for MN differentiation. Six days later, MN aggregates were dissociated with Accumax, and SC were detached using Accutase. MN and SC were then plated at a 1:1 well ratio onto C2C12, either individually for cocultures or combined for tricultures. Cells were plated in triculture medium (triM) consisting of MN basal medium supplemented with 0.1 mM ascorbic acid, 0.1 µM purmorphamine, 0.5 µM retinoic acid, 10 ng/mL BDNF, 10 ng/mL GDNF, 10 ng/mL IGF-1, and 200 ng/mL NRG1. Next, 10 µM Y-27632 was added upon seeding, and 24 h later, medium was changed to triM without Y-27632. At all following medium changes, only half the volume of medium per well was changed in order to cause minimal disturbance to cells.

### 2.6. Immunofluorescence Staining and Microscopy

Cultures were fixed with 4% PFA for 15 min, washed in PBS 3 x, and either stored at 4 °C until staining or stained immediately. For stainings of SC including SOX10, cells were permeabilized with 0.1% Triton X-100 for 3 min and then incubated for 1 h at RT in blocking buffer (2% BSA in PBS). BSA blocking buffer was used subsequently for washes and antibody incubations. For all other stainings, cultures were first incubated for 10 min at RT in a blocking buffer containing 0.1% saponin and 0.2% fish skin gelatin in PBS, which was also used for washes. Antibodies were incubated in 0.01% saponin/0.2% gelatin in PBS. For both protocols, primary antibodies were incubated for 1 h at RT, and secondary antibodies were incubated for 45 min at RT in the dark. AChR were labelled with α-bungarotoxin (αBTX) Alexa Fluor 647 conjugate (Thermo Fisher Scientific) incubated together with secondary antibodies. For a list of all primary antibodies used, see Appendix A. Imaging for differentiation marker analysis in SC was done using a Leica Aperio Versa Slide Scanner (Leica Biosystems, Wetzlar, Germany) with HC PL FLUOTAR 20×/0.55 DRY objective and DAPI, Cy5, and Spectrum Green filters. All other imaging was performed using an inverted Leica TCS SP8 confocal microscope (Leica Microsystems, Wetzlar, Germany) with HC PL APO 20×/0.75 IMM CORR CS2 objective and 405 nm, 488 nm, 561 nm, and 633 nm lasers and Leica Application Suite X software. For myotube morphology analysis, image data obtained with both platforms were used. To collect data from most of the culture area while excluding coverslip edges, a 7.5 × 7.5 mm^2^ square region central on Ø 12 mm glass coverslips was imaged for each condition. For stainings designated for quantitative AChR analysis, single optical planes determined by autofocus in the αBTX staining channel were imaged at random regions on the coverslip.

### 2.7. Image Segmentation and Quantitative Analysis

Quantification of marker proteins stained in MN and SC differentiations was performed by detection of cell nuclei (visualized with DAPI) and subsequent measurement of signal intensity in this region for the stained marker in a second channel. For quantification of nuclear stainings of SOX10 and Ki67, detection of nuclei was performed using the Bead-Net software [27]. A manually labeled dataset of 31 images (128 × 128 px^2^) was created for training and testing the algorithm. The detected nuclei center points were then used to determine whether the cell was positive or negative for the analyzed staining. To achieve this, the marker staining image was smoothed by a Gaussian filter with sigma = 0.5. Then, the maximum value inside a 3 × 3 px^2^ region, centered at the detected nuclei location, was calculated and compared to a user-defined threshold. A maximum value greater than the threshold was counted as a positive cell; otherwise, the cell was counted as negative. For quantification of S100b and P0, the individual nuclei were segmented using the TWANG algorithm [28] implemented in XPIWIT [29], and mean intensity values obtained from the marker staining channel within the segmented nuclear area were compared to a user-defined threshold to determine positive cells.

To give an indication of how many elongated bipolar cells were present and how pronounced their structure was, the elongation index was introduced. This measure provides information about the summed length of narrow structures. To calculate this value, images of SC cultures stained for S100b were smoothened by a Gaussian filter with sigma = 8 and segmented using a user-defined threshold. Subsequently, narrow structures were removed from the segmented image by performing 8 iterations of binary erosion followed by 8 iterations of binary dilation with a 3 × 3 px^2^ diamond-shaped structure element. To calculate the difference between the processed and raw segmentation, an element-wise XOR operation was performed. The resulting segmentation showed mainly narrow structures. To remove artifacts, objects with an area smaller than 960 px^2^, an eccentricity smaller than 0.75, or a major axis length smaller than 120 px were deleted. Finally, the elongation index was calculated by adding up the major axis lengths of the remaining objects and dividing it by the number of positive cells. Cell classification and calculation of the elongation index were performed in Python.

In tricultures, quantification of SC alignment, as well as AChR cluster colocalization, was performed by manual counting and classification using confocal image stacks. For myotube area analysis, images were analyzed using labels allowing the visualization of myotubes (i.e., MF20, desmin, or αBTX). Myotubes were manually outlined and their individual areas obtained using the Fiji distribution of ImageJ [30]. Data were then processed in R (see Statistical Analysis). For AChR cluster segmentation, confocal single plane images were checked manually, and images with obvious detachment of myotubes were excluded. Image analysis was done in Fiji with a custom macro to semi-automate the process. For each experimental condition, a coverslip area of at least 25 mm^2^ was analyzed. Fluorescent clusters were segmented by thresholding and manual selection. For each segmented cluster, area and mean intensity were measured. Subsequently, whole myotubes were visualized by increasing image contrast and segmented, and the area was measured for normalization of cluster number.

### 2.8. Statistical Analysis

Results are presented as mean ± SD. Statistical analysis was performed using the software GraphPad Prism 8. At least 3 independent experiments were analyzed for each parameter. Data were tested for normal distribution by the Kolmogorov–Smirnov test. If not stated otherwise in the following, results were analyzed using one-way ANOVA with Tukey’s multiple comparisons test. Comparison of SOX10+ cells shown in Figure 3 was performed with paired t-tests. Statistical analyses shown in Figure 6 were performed in R, version 4.0.5 (R Core Team, 2021). The dplyr package [31] was used for data manipulation; figures were produced using the packages ggplot2 [32] and scales [33]. In addition, lme4 [34] was used to produce a linear mixed-effect model. Furthermore, *p*-values less than 0.05 were considered significant and reported as * *p* < 0.05, ** *p* < 0.01, *** *p* < 0.001. Figures were prepared in Adobe Illustrator CC 2018.

## 3. Results

### 3.1. Differentiation of Motoneurons (MN) and Schwann Cells (SC) from hiPSC

For differentiation of MN from hiPSC, a small molecule differentiation protocol described by Du and colleagues [26,35] was employed with slight modifications (see Methods section). Images of differentiation stages and the sequence of molecular factors are shown in Figure 2A. Briefly, hiPSC were induced to neuroepithelium for six days by dual SMAD inhibition and WNT activation followed by caudal and ventral patterning with retinoic acid and the Sonic hedgehog activator purmorphamine. After 12 days, MN precursors positive for Olig2 (71.5 ± 11.4% of total cells, mean ± SD) were obtained (Figure 2B,C). These could be expanded and cryopreserved. From day 12 on, precursors were cultured in suspension with added neurotrophic factors for six days and then seeded adherently again for final maturation. MN obtained after 20–30 days of differentiation were positive for vesicular acetylcholine transporter (vAChT; 93.0 ± 2.9% of total cells), as shown in Figure 2B,C. MN could be cultured and further matured up to day 50 and were furthermore confirmed by immunostaining to express βIII-tubulin (Tubb3), Hb9, Isl1, choline acetyltransferase (ChAT), Map2, tau, peripherin (Prph), and synaptophysin (Sph); representative images are shown in Figure 2D and Appendix A.

Differentiation of SC was based on a protocol by Kim and colleagues [24] which yielded SC precursor cells within 24 days (Figure 3A). Accordingly, cells were neuralized by activin inhibition and WNT activation but without inhibiting BMP. From day 6 on, neuregulin-1 (NRG1) was added to the medium to induce neural crest differentiation towards SC fate. To further promote neural crest specification, we additionally included BMP4 and the BMP inhibitor DMH1 from days 6 to 12. This aspect was not part of the original protocol from Kim et al. and is described in more detail in Section 3.2. Subsequently, SC precursors were obtained after 24 days and differentiated to immature SC through increased concentration of NRG1 and added factors PDGF-BB, forskolin, and retinoic acid. From day 30 onwards, SC were cultured in their final differentiation medium containing NRG1 and FBS and then expanded and cryopreserved.

### 3.2. Tuning of BMP Activation Improves Robustness of Schwann Cell Differentiation

While SC positive for S100b could be generated with the protocol described above, differentiation success and yield across differentiation batches were both quite variable. Thus, as suggested recently by Hackland and colleagues [36], we tested whether precise intermediate activation of BMP signaling by using a small molecule top-down inhibition system was able to increase protocol robustness with respect to promoting neural crest cell derivation. To this end, a saturating concentration of exogenous BMP4 was applied, while at the same time, BMP signaling was attenuated using the BMP inhibitor DMH1 to achieve optimal BMP activity independent from fluctuations in endogenous BMP levels. Based on the work of Hackland et al., we used the same concentrations of BMP4 and DMH1 and a similar treatment time, which started following the first six days of neuralization at the same time as NRG1 was added to induce specification of neural crest cells. After 12, 19, and 26 days of differentiation, the cultures were screened for the presence of the neural crest marker SOX10 (Figure 3B). At all timepoints evaluated, the mean proportion of SOX10+ cells increased significantly upon activation of BMP signaling; in addition, yields of SOX10+ cells were much more consistent between different experiments with tuned BMP as compared to cells not treated with BMP4 and DMH1 (Figure 3C; percentage of SOX10+ cells increased from 21.0 ± 23.5% to 62.6 ± 14.1% at day 12, from 46.6 ± 28.4% to 90.2 ± 10.0% at day 19, and from 45.1 ± 30.3% to 77.0 ± 11.3% at day 26; mean ± SD). Confirming earlier reports [36], the beneficial effect of BMP4 addition was only observed if the BMP antagonist DMH1 was applied simultaneously (Appendix A). SOX10+ cells derived with this modified protocol were successfully differentiated further into SC positive for SC markers, myelin protein zero (P0) and S100b (Figure 3D). Furthermore, we tested whether the protocol could work in completely defined conditions, i.e., by replacing the ill-defined biologicals, BSA and FBS, with specific molecular components. This showed that addition of FBS from day 24 onwards was necessary for a sufficient yield of Schwann cells. In serum-free conditions, Schwann cells were able to differentiate but at low survival and efficiency (18.7 ± 5.7% SOX10+ cells, 20.6 ± 6.3% positive for S100b at day 38; Appendix A). However, BSA and β-mercaptoethanol were dispensable during SC precursor differentiation, regardless of whether BMP4 and DMH1 were added or not (Appendix A).

### 3.3. Differentiation of hiPSC-Derived Schwann Cells Is Promoted in Triculture Medium

In preparation for triculture establishment, we tested whether differentiated SC were supported in a medium formulation designed for tricultures based on MN differentiation medium. For simplicity, we refer to this medium as “triculture medium” (triM). To support the different cell types in tricultures, triM contained differentiation and maturation factors NRG1, retinoic acid (RA), purmorphamine (Pur), ascorbic acid (AA), BDNF, GDNF, and IGF-1. After switching hiPSC-SC monocultures to triM, we observed a rapid change in cell morphology towards a bipolar elongated spindle shape, as would be expected for well-differentiated SC (Figure 4A). To identify individual molecular factors affecting these morphological changes, cells were cultured in either standard SC medium (SCM), fully supplemented triculture medium (triM), triculture base medium (BM), BM supplemented with NRG1 only, BM supplemented with NRG1, RA, and Pur, or BM supplemented with NRG1, RA, and AA. After three days of culture, cells were fixed and immunostained for differentiation and maturation markers (Figure 4A; Appendix A shows all conditions). The percentage of cells positive for Ki67, P0, and S100b, respectively, was quantified by image segmentation (Figure 4B–D). After three days in triM, a drop of Ki67+ cells to 18.1 ± 5.9% (mean ± SD), compared to 38.0 ± 6.3% in SCM, indicated a decreased proliferation in triM. Under the same condition, expression of SC markers increased, i.e., from 53.6 ± 15.1% in SCM to 83.4 ± 11.1% in triM of cells positive for S100b and from 27.1 ± 17.9% in SCM to 80.9 ± 13.4% in triM of cells positive for P0. Conversely, BM for three days led to no significant changes (21.8 ± 7.9% Ki67+; 63.7 ± 12.6% S100b+; and 56.0 ± 13.5 % P0+) and nor did BM supplemented with NRG1 only (23.5 ± 14.8 % Ki67+; 69.5 ± 16.3% S100b+; and 55.6 ± 6.7% P0+). In contrast, upon supplementation of BM with NRG1, RA, and Pur, the percentage of Ki67+ cells was reduced significantly to 15.0 ± 10.9%, and that of P0+ cells significantly increased to 82.9 ± 6.4%, while the proportion of S100b+ cells showed an upward trend which was not significant (78.4 ± 14.6%). Similarly, in BM supplemented with NRG1, RA, and AA, Ki67+ cells decreased to 15.1 ± 9.2%, and S100b+ and P0+ cells increased to 88.6 ± 6.6% and 82.9 ± 9.1%, respectively. In summary, these data suggested that RA and Pur/AA synergistically reduced proliferation and enhanced SC marker expression during the SC maturation period.

To further consolidate this finding, we determined the occurrence of spindle-shaped cell morphologies as a proxy for SC maturation. To this end, a segmentation analysis pipeline was developed which yielded a numerical value termed “elongation index”. This positively correlated with the presence of thin, elongated cellular processes. Therefore, fluorescence images of cell cultures stained for S100b were first segmented by manual thresholding. Subsequently, all narrow structures were removed from the segmented image by erosion and dilation operations (for detailed description, see Methods section). By subtracting the resulting images from the originally segmented images, only narrow structures remained (Figure 4E). For each condition, the major axis lengths of these structures were added up and divided by the respective number of S100b+ cells to obtain the elongation index (Figure 4F). For cells cultured for three days in either SCM or triM, this method retrieved elongation indices of 5.3 ± 5.0 and 23.9 ± 9.3, respectively (Figure 4F). Consistently, while no significant changes of elongation index were observed for BM and BM + NRG1 cultures (8.6 ± 6.1 and 6.7 ± 4.5, respectively), BM + NRG1 + RA + Pur (19.3 ± 7.0) and, even more pronounced, BM + NRG1 + RA + AA (31.2 ± 9.7) led to clearly enhanced elongation. In conclusion, differentiation of hiPSC-derived SC was promoted in triM, as illustrated by an increase in S100b- and P0-positive cells, a decrease in proliferation, and morphological changes towards a more elongated cell shape; according to our data, RA and AA were major drivers of these improvements.

### 3.4. Tricultures Including hiPSC-Derived SC Can Be Prepared from Frozen Cells in Nine Days

As mentioned, triM was based on the final MN differentiation medium, but Compound E was replaced with NRG1 to support SC. When used instead of standard C2C12 differentiation medium (DM), the serum-free triM formulation was able to induce differentiation of C2C12 into myotubes which showed spontaneous contractions. However, compared to DM, differentiation seemed to progress more rapidly in triM, and already after few days, extensive cell detachment and cell death was observed. This effect could be prevented if differentiation was induced initially by culturing C2C12 in DM and switching to triM subsequently. Therefore, in order to create tricultures C2C12 cells were seeded first and left for 24 h in differentiation medium. Then, hiPSC-derived SC and MN were added and the cultures switched to triM. This way, starting from cryopreserved C2C12 myoblasts, MN precursors, and SC, tricultures were obtained in nine days (Figure 5A). All three cell types were thawed on the same day in their respective media. C2C12 and SC were regularly passaged until tricultures were established. MN precursors were grown to a dense cell layer and subsequently cultured in suspension for six days. Two days prior to triculture seeding, C2C12 myoblasts were plated and differentiation to myotubes was induced 24 h later by switching to myoblast differentiation medium. Another 24 h later, the neurospheres obtained during the suspension phase were dissociated into a single-cell suspension and added to muscle cultures together with SC. Cultures usually developed within a few days to show contracting myotubes. Before massive contraction-induced detachment of the cultures from the glass surface, tricultures were fixed for evaluation three days after seeding of neurons and SC. Subsequently, staining for specific marker proteins was performed. These included Tubb3, synaptophysin, and vAChT for MN; S100b and vimentin for SC; and, finally, desmin or myosin heavy chain for myotubes. Additionally, nicotinic acetylcholine receptors (AChR), a marker for NMJs, were labelled with the snake venom α-bungarotoxin (αBTX) coupled to a fluorescent dye. Frequently, an alignment of SC with neurites was seen (79.9 ± 8.3% of SC in tricultures were completely or partially aligned with neurons; mean ± SD, n = 4) as well as colocalization of AChR with MN (70.0 ± 6.6% of all AChR clusters in tricultures colocalized with MN) or with both MN and SC (17.6 ± 10.1% of all AChR clusters; mean ± SD, n = 4). Representative confocal microscopy images of tricultures are shown in Figure 5B and Appendix A. To test whether the protocol could potentially be adapted to a completely hiPSC-derived triculture system, we also generated myotubes from hiPSC [37] and successfully cocultured them with hiPSC-derived MN and SC (Appendix A).

### 3.5. Cocultures Promote Formation of Myotubes with Increased Cytoplasmic Area

Next, the effects of cocultures on myotube differentiation were addressed. To this end, C2C12 cells were kept in either monoculture (C2C12 only), biculture (C2C12 combined with either SC or MN), or triculture (C2C12, SC, and MN); all conditions were cultured in triM. Cultures were immunostained against myosin heavy chain to visualize differentiated myotubes (MF20, Figure 6A). The microscopic assessment of myotubes led to the identification of two myotube populations: (i) myotubes with a large cytoplasmic area (>200,000 µm^2^) and (ii) myotubes with a smaller cytoplasmic area (<200,000 µm^2^). For both myotube populations, the mean percentage of coverage compared to the total area of the analyzed picture was determined for each culture condition. This showed that, starting with a value of 1.97 ± 4.31% in C2C12 monoculture (mean ± SD), the large-type myotubes increased the coverage upon biculture to 3.80 ± 5.37% and 6.88 ± 12.56% for C2C12 + SC and C2C12 + MN bicultures, respectively. In the triculture condition, the large-type myotube coverage was further enhanced to 9.52 ± 17.00% (Figure 6B). Conversely, coverage of myotubes with small cytoplasmic area was decreased in all coculture conditions compared to the C2C12 monoculture condition. Differences were significant for the biculture with MN and the triculture condition (4.31 ± 0.02% for C2C12 monocultures; 0.01 ± 0.03% for C2C12 + SC; 0.01 ± 0.02% for C2C12 + MN; and 0.009 ± 0.02% for tricultures) (Figure 6C). While the division of the myotube populations according to their area was preserved in the different coculture conditions, a progressive enlargement of the density curve for the large-type myotube population (Figure 6D, pink curves) was observed. This enlargement was slight in the biculture with SC, was more pronounced in the biculture with MN, and was even stronger in the tricultures (Figure 6D). All in all, these results indicated that the presence of SC and MN influence myogenesis in a synergistic manner.

### 3.6. hiPSC-Derived MN and SC Influence Clustering of AChR on Myotubes In Vitro

To investigate clustering of postsynaptic AChR on myotubes, AChR were stained with fluorescently labelled αBTX. Stained receptor clusters on myotubes were visualized by confocal fluorescence microscopy, segmented, and analyzed for their area, fluorescence intensity, and number. Results were compared for cultures of myotubes alone, C2C12 bicultures with MN or SC, and tricultures. Figure 7A shows stainings for specific markers of the individual cell types in the different cocultures as well as representative segmented AChR clusters. Quantitative analysis of AChR mean cluster sizes revealed a significant reduction from 142.7 ± 48.6 µm^2^ in C2C12 monocultures to 19.6 ± 1.7 µm^2^ and 22.4 ± 5.0 µm^2^ (mean ± SD) in C2C12 + MN and tricultures, respectively. In bicultures with SC, a trend towards decreased mean area (90.5 ± 31.3 µm^2^) was seen as well, but this was not statistically significant (Figure 7B). Regarding mean fluorescence intensity per cluster, an upward trend was found in cocultures when compared to myotubes alone, but the differences were not significant (Figure 7C). Finally, the number of segmented AChR clusters normalized to myotube area was analyzed. Compared to 16.2 ± 15 segmented clusters per mm^2^ myotube area in C2C12 monocultures, the number of receptor clusters increased significantly in cocultures of myotubes and MN (112.8 ± 9.0). In tricultures, the number of clusters was also increased but to a lesser extent than in bicultures with MN (82.0 ± 14.3 per mm^2^ in tricultures). In bicultures with Schwann cells, mean number per mm^2^ was 23.1 ± 14.9, showing no significant difference to C2C12 monoculture (Figure 7D). Additionally, myotubes were double-stained for the protein rapsyn, which is crucial for nerve-induced AChR clustering during muscle development (reviewed in [38]). In C2C12 monocultures, only 36.3 ± 3.1% of segmented AChR clusters stained positive for rapsyn (Appendix A). This value increased significantly in biculture with MN (58.0 ± 5.5%) and in tricultures (60.2 ± 1.9%). In bicultures with SC, 48.6 ± 12.6% of segmented clusters were positive for rapsyn, showing a non-significant increasing trend (Appendix A). Taken together, these results confirm the inducing effect of hiPSC-derived MN on AChR cluster formation and suggest an additional influence of SC coculture.

## 4. Discussion

At the developing NMJ, SC are crucial for the formation of mature and stable synaptic contacts [7,8,39]. Even beyond synaptogenesis, SC remain an essential component of NMJs, regulating NMJ structure, function, and maintenance [11,13,40,41] as well as NMJ plasticity and remodeling upon injury [14,42,43,44]. Despite being indispensable for NMJs in vivo, their integration into in vitro NMJ models has been recognized as an important feature [1,3,6,45], but it was hampered by experimental pitfalls. Our current knowledge of SC biology is primarily obtained from animal models, and while some mechanisms translate from rodents to human biology, recent studies have revealed significant species differences of NMJ components [46] and, specifically, SC [47]. In the light of emerging roles of SC not only in synaptogenesis but also in neuromuscular diseases, the need for models with a human genetic background, which also include SC, is particularly relevant.

For example, in several SOD1 mutant mouse models, which are widely used to study pathology of amyotrophic lateral sclerosis (ALS), alterations in SC activity and morphology precede denervation of NMJs and impede NMJ repair [48,49,50]. Interestingly, altered SC morphology is also seen in ALS patients [51,52]. However, species-specific differences complicate translation of disease signs and mechanisms. In comparison to mice, SC at healthy human NMJs exhibit higher non-synaptic placement of their nuclei along with less AChR coverage, which in SOD1 mutant mice would both be associated with pathological changes [53]. Indeed, mouse model data have so far not been able to translate well into treatments for human ALS [54]. Therefore, integration of human SC would likely advance the relevance of stem-cell based models not only for studying cell type interactions at the healthy NMJ but also for disease modeling and, ultimately, drug testing. So far, selective integration of human SC into neuromuscular cocultures has been impaired by several factors: (i) introduction of a third cell type in defined cocultures considerably increases complexity of the culture model [2]; (ii) protocols to derive SC from hiPSC are sparse, often coming with low yields, purification steps, and long differentiation times [21]; and (iii) differentiation protocols to specifically derive terminal SC are not available, since exact developmental switches necessary for their specification are elusive [55]. In this work, we provide a protocol to set up tricultures comprised of hiPSC-derived SC, MN, and C2C12 myotubes in only nine days from frozen cell stocks.

To achieve this, we present an improved differentiation protocol to derive human SC from hiPSC with increased robustness. When employing a SC differentiation protocol which does not necessitate purification steps [24], a secondary cell population positive for SOX2 but negative for SOX10 was frequently observed, suggesting insufficient neural crest induction. This was in accordance with literature, where oftentimes low cell yields are reported in differentiation protocols for obtaining neural crest in general and, specifically, SC [21,36]. A possible reason could be insufficient activation of BMP signaling, which is required for neural crest specification [56]. However, deliberate BMP activation by small molecules can also antagonize neural crest specification [57], and timing regarding the differentiation process is crucial [58]. The work of Hackland et al. proved that precise activation of BMP signaling to an intermediate level is necessary for the development of neural crest cells and demonstrated the presence of different levels of endogenous BMP between different PSC lines and experiments. Their findings explain previous conflicting results regarding either promotion or inhibition of neural crest specification upon BMP activation in cell culture and could also be a reason for low cell yields or varying differentiation success between distinct PSC lines and differentiation batches in SC differentiation protocols. In such protocols, which do not add compounds to interfere with BMP signaling, differentiation efficiency would possibly depend on inherent BMP signaling levels in PSC lines and culture systems used. Consequently, we explored a top-down inhibition system as suggested by Hackland et al., i.e., saturating with BMP4 exogenously while using the BMP antagonist DMH1 to attenuate signaling to a precise level. In accordance with their results, efficient and robust derivation of SOX10+ neural crest cells could be achieved, and these cells were found to be competent to further differentiate along the SC lineage.

Since non-defined media constituents such as serum are subject to batch-to-batch variation and may contain components which affect BMP signaling or other relevant pathways, thereby possibly increasing protocol variability, we explored xeno-free medium formulations for differentiation. SC precursor cells derived after 24 days of differentiation were obtained in completely defined medium conditions without addition of BSA, which is frequently used in this step. Usage of BSA can potentially affect signaling during differentiation, since it acts as a carrier protein and might be bound to molecules influencing signaling targets or activity of small molecules added for differentiation [36,59]. Indeed, we obtained SC precursors using a concentration of Activin/Nodal/TGF-β pathway inhibitor SB431542 reported for stem cell neuralization and also neural crest specification in chemically defined conditions [36,60]. Conversely, protocols involving BSA often require 10-fold higher concentrations of that molecule [24,57,61]. Thus, while SC precursor derivation could be obtained in a xeno-free manner, addition of FBS following day 24 of differentiation was necessary to mature SC precursors into SC without contaminating cell populations. Although in serum-free medium containing N2 and SM1 supplements, a population of differentiating SC expressing S100b was observed as well, and cell yields and differentiation efficiency were drastically decreased. Potentially, the present protocol could be adapted to be serum-free by introducing a cell-sorting step. This would render it suitable for applications requiring the use of defined xeno-free media throughout the whole differentiation period, such as for clinical tissue engineering. Even if experimental design allows the use of serum for differentiation of cells, it would be desirable to perform subsequent experiments in defined, serum-free conditions to avoid variable and hard to predict interferences from such compounds. The medium formulation here presented for the culture of NMJ tricultures fulfilled these requirements and also supported hiPSC-derived SC in monoculture after a preceding maturation phase in serum-containing medium.

Moreover, switching SC to the serum-free triculture medium (triM) seemed to improve cell differentiation, and possibly maturation, after only three days of culture, as determined by their more elongated, characteristic spindle-like morphology, their increase in expression of the SC markers myelin protein zero (P0) and S100b, and a decrease in proliferation. Our results suggest that this effect was mainly mediated by retinoic acid (RA), since proliferation was decreased and P0 expression as well as cell elongation were increased in all conditions containing this factor. RA is a crucial patterning and maturation factor in neural development and drives posterior specification of trunk identity in neural crest differentiation [62,63]. Furthermore, it is included in a number of differentiation protocols which generate SC or Schwann-like cells from human stem cells or other cell sources, including mesenchymal stem cells, adipose-derived stem cells, or fibroblasts [24,64,65,66]. SC express retinoid receptors during development and in adult peripheral nerve, and RA can exert different functions on SC, suggesting a key role for retinoic signaling in SC physiology [67]. Moreover, RA appeared to act in synergy with ascorbic acid. Ascorbic acid is known to promote SC basal lamina assembly and myelin formation [68,69], and it is frequently added in cocultures of SC with primary neurons to induce myelination [22,23,70]. This induction of the myelination program is accompanied by a shifting to a differentiated postmitotic state and upregulation of protein expression related to myelination, including P0 [71,72], but all exact mechanisms of action have not been elucidated yet [68]. RA, on the other hand, was found to inhibit myelination in primary explant cultures of mouse and rat peripheral nerve [73]. Nevertheless, this mechanism was accompanied by increased expression of P0, which is in accordance with our findings. Further studies in the same system also revealed an inhibitory effect of RA on SC proliferation [74]. However, the mechanism downstream of this antiproliferative and potentially maturating effect has yet to be elucidated, as well as other possibly co-regulated pathways. Altogether, our findings provide valuable benefits to protocols for differentiation of SC from hiPSC currently available in the literature. Further experiments would be advisable to find the optimal medium formulation for hiPSC-derived SC maturation if NMJ tricultures are not the primary experimental goal.

For our study, these hiPSC-derived SC were used to set up tricultures in combination with hiPSC-derived MN and C2C12 myotubes. While the interplay between muscle cells, SC, and MN is known to be crucial to form and mature functional NMJs, the importance of their crosstalk on differentiating muscle cell homeostasis is still puzzling. In the current work, the presence of SC and MN was demonstrated to influence myogenesis in a synergistic manner. This finding may be explained by the secretory activities of MN and SC that trigger different biological processes. Indeed, SC can secrete several different neurotrophins, and Kim et al. demonstrated expression and secretion of BDNF, GDNF, NGF, and NT-3 by hiPSC-derived SC. These are known to exert pleiotropic effects on diverse cell types and tissues. Amongst the targeted tissues, skeletal muscle cells express receptors for various growth factors, suggesting that neurotrophic signaling could play a crucial role in the myogenic process [75,76,77,78]. While the lack of NT-3 or its receptor, TrkC, considerably alter mobility [79,80], the BDNF pathway was highlighted as a major factor in skeletal muscle regeneration and myogenesis [81,82,83]. BDNF may act as an autocrine/paracrine factor that transcriptionally regulates metabolism in muscle cells [84]. Additionally, NGF was described to regulate cellular homeostasis and inflammatory status in skeletal muscle fibers [85,86]. Similarly, MN are also known to release important paracrine factors that modulate muscle cell physiology. Amongst these, agrin is a key factor for clustering AChR at developing NMJs via the agrin/Lrp4/Musk pathway [87,88]. It can also induce rapsyn stabilization at AChR clusters [89,90,91,92]. In alignment with that, we found increased percentages of AChR clusters positive for rapsyn in myotubes cocultured with MN. In bicultures of myotubes and SC, trends towards increased numbers of AChR clusters and of rapsyn-positive clusters were observed. Although these changes were not statistically significant, they were in the same order of magnitude (1.42-fold and 1.34-fold increase for number of clusters and percentage of rapsyn-positive clusters, respectively) and were accompanied by a general change of appearance of AChR clusters; while some of them resembled the larger and more diffuse clusters observed in C2C12 monocultures, others were rather similar to the smaller and more defined clusters found in cocultures with MN. In fact, primary SC at the NMJ were reported to express an isoform of agrin which is able to enhance AChR clustering, although with lower efficiency than neural agrin [7,93,94,95].

Next, in tricultures, the number of AChR clusters was decreased when compared to bicultures with only MN (0.73-fold decrease). Considering that SC are coordinators of synapse elimination in NMJ development and they are able to selectively stabilize or destabilize synaptic contacts to achieve matured innervation [10,96,97], this finding would be potentially interesting. One possible mode of action for the loss of AChR clusters could be a direct influence of SC on MN-derived agrin levels. At NMJs, matrix metalloproteinase-3 secreted by terminal SCs was found to cleave agrin and reduce its concentration in the synaptic basal lamina [98,99,100]. Conversely, AChR aggregates were increased in matrix metalloproteinase-3 null mice [101,102]. Moreover, experiments in chick and *Xenopus* myotubes showed that several trophic factors, some of which have been shown to be expressed by hiPSC-derived SC [24], can lead to suppression of AChR clustering via inhibition of agrin synthesis and enhanced transmitter release in MN [103,104]. In development, AChR expression, clustering, and stabilization are regulated positively by agrin and negatively by ACh. While ACh exerts its effects globally, agrin acts locally to stabilize only directly innervated AChR clusters, thus leading to a destabilization of aneural clusters [87,105,106]. An increase in ACh release by MN induced by SC-secreted neurotrophic factors could, therefore, lead to destabilization of AChR clusters not in direct contact with neurites and, hence, account for the decreased number of AChR clusters in tricultures.

The protocol to set up tricultures presented in this work employed the straightforward two-layer 2D culturing method that has been widely used for neuromuscular cultures. This makes it flexible and easy to implement with already established in vitro NMJ models accessible for analysis via imaging and molecular assays. Other authors have reported functionality of in vitro NMJs from different cell sources [107,108,109]. This was done by evaluating muscle contraction upon excitation of MN, either by glutamate [110,111] or optogenetic [112,113] or electrophysiological [108,114] stimulation and by testing the specificity of transmission response by antagonists which either blocked ACh release or binding [16,107,115]. In this context, it would be informative to integrate hiPSC-derived SC into NMJ culture systems suited for such readouts in order to evaluate a possible impact on in vitro NMJ functionality. Moreover, besides functionality, it would be of interest whether hiPSC-derived SC are able to increase maturity of in vitro NMJs. One method to judge maturation is to evaluate the switch of the embryonic gamma subunit to the adult epsilon subunit in AChR; however, in vitro, this has been demonstrated so far only in advanced explant or engineered 3D cultures [16,111,112]. Recent work has shown enormous benefit of 3D culture methods on stability, maturity, and organization of neuromuscular culture models and demonstrated their potential for disease modeling [107,111,113,116]. Since myotubes tend to detach from rigid culture substrates over time and are, therefore, not suited for long-term experiments in 2D cultures [117,118], long-term studies concerning in vitro NMJ maturation and stabilization would require such 3D culture approaches. Therefore, to explore its full potential, the triculture model should be ultimately advanced to a suited 3D culture platform. Such cultures could potentially be used for studying early NMJ development in vitro and the contribution of the distinct cellular partners to synapse formation and stabilization as well as the effect of the presence of SC on differentiation of the other cell types and vice versa. In stable long-term models, SC differentiation could be further advanced and explored, for example, by checking for distinct specification into myelinating and non-myelinating subtypes or even possible SC responses upon modeling injuries which are seen in vivo [44,119]. Furthermore, the use of hiPSC opens the possibility for cell type-targeted disease modeling. Besides comparing cultures completely derived from patient hiPSC with control cultures, it would be possible to create models in which only one of the three cell types is derived from patient cells in order to elucidate cell type-specific contributions to disease mechanisms. Along these lines, we have demonstrated the possibility to set up tricultures from hiPSC-derived cells only, including hiPSC-derived muscle cells as well, and future work will focus on robust protocols to create isogenic human tricultures and transferring them to 3D.

## 5. Conclusions

The presented triculture model is the first to show selective and controlled integration of SC which were separately differentiated from hiPSC with improved robustness into neuromuscular cocultures. It paves the way for further studies and more complex models exploring their role in maturation and pathological mechanisms at the NMJ. Adapting the protocol to more advanced 3D culture systems to enable long-term studies will open possibilities for studying the influence of hiPSC-derived SC on in vitro NMJ formation and maturation and also the cues potentially provided by the other cells which affect SC differentiation and their specification to a terminal SC fate.

## Figures and Tables

**Figure 1 cells-10-03292-f001:**
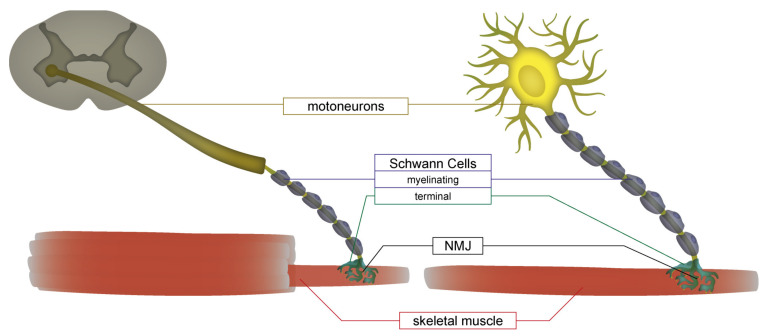
Scheme of the cell components of the core NMJ. Cell bodies of lower motoneurons are located in the ventral horn of the spinal cord (gray structure in the upper left corner), from which their axons extend through spinal nerves to skeletal muscles. Each muscle fiber is innervated at a single synaptic connection, the neuromuscular junction (NMJ). Terminal Schwann cells form an integral part of these synapses; they are different from myelinating Schwann cells, which wrap the axons of motoneurons. For simplicity, further cell types at the NMJ, such as kranocytes and sympathetic neurons, were left out.

**Figure 2 cells-10-03292-f002:**
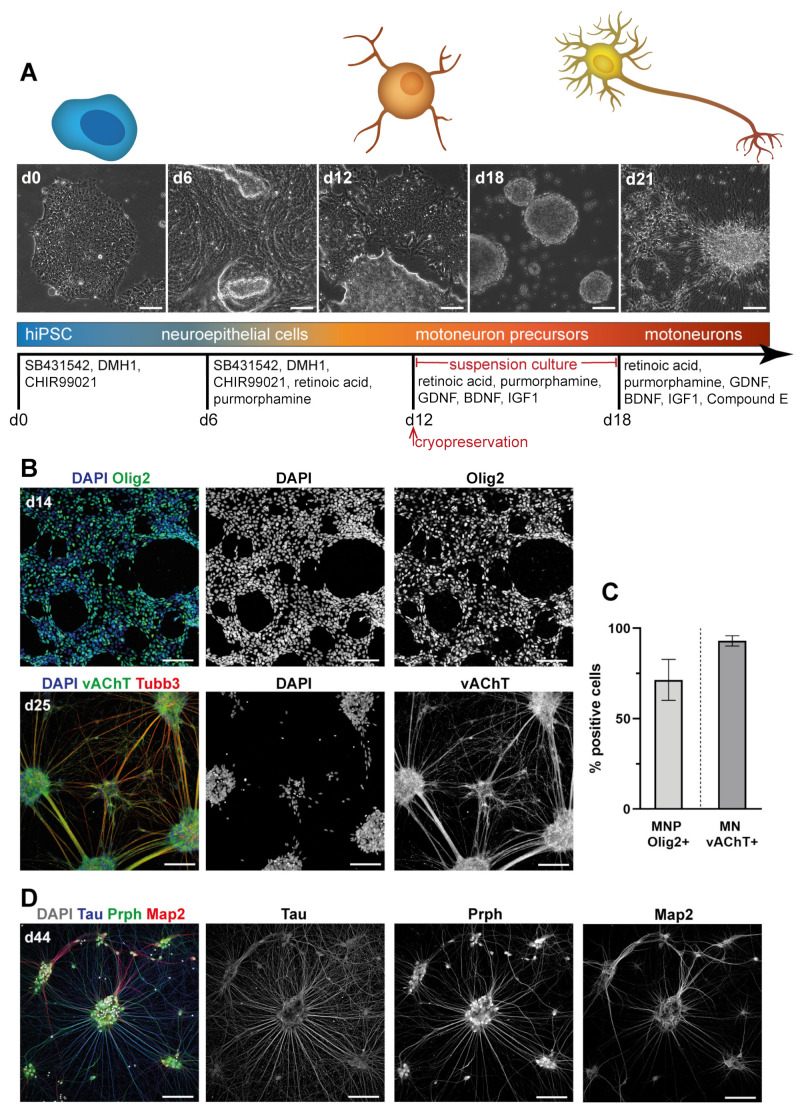
Morphological characteristics and marker protein expression indicate differentiation of motoneurons from hiPSC. (**A**) Schematic representation of protocol timeline and differentiation factors used. Micrographs show representative brightfield images of cell populations at differentiation times as indicated on the top left of images. MN precursors (MNP) could be cryopreserved after 12 days; MN were obtained after 20–30 days of differentiation. Scale bars: 100 µm. (**B**–**D**) hiPSC were differentiated according to the protocol shown in A, fixed at time points as indicated on the top left of panels, and then immunostained for markers as displayed above the panels. Nuclei were labeled with DAPI. (**B**) Representative micrographs with fluorescence signals for Olig2 in MNP (upper panels) and vesicular acetylcholine transporter (vAChT)/βIII-tubulin (Tubb3) in MN (lower panels). Color codes in merge images as indicated. Scale bars: 100 µm. (**C**) Quantification of Olig2+ and vAChT+ cells at d14 (n = 5 experiments) and d25 (n = 4 experiments), respectively. All data were normalized to total cell count. Graph depicts mean ± SD. (**D**) Mature hiPSC-derived MN stained for Tau, Map2, and Peripherin (Prph). Color code in merge image as indicated. Scale bars: 100 µm.

**Figure 3 cells-10-03292-f003:**
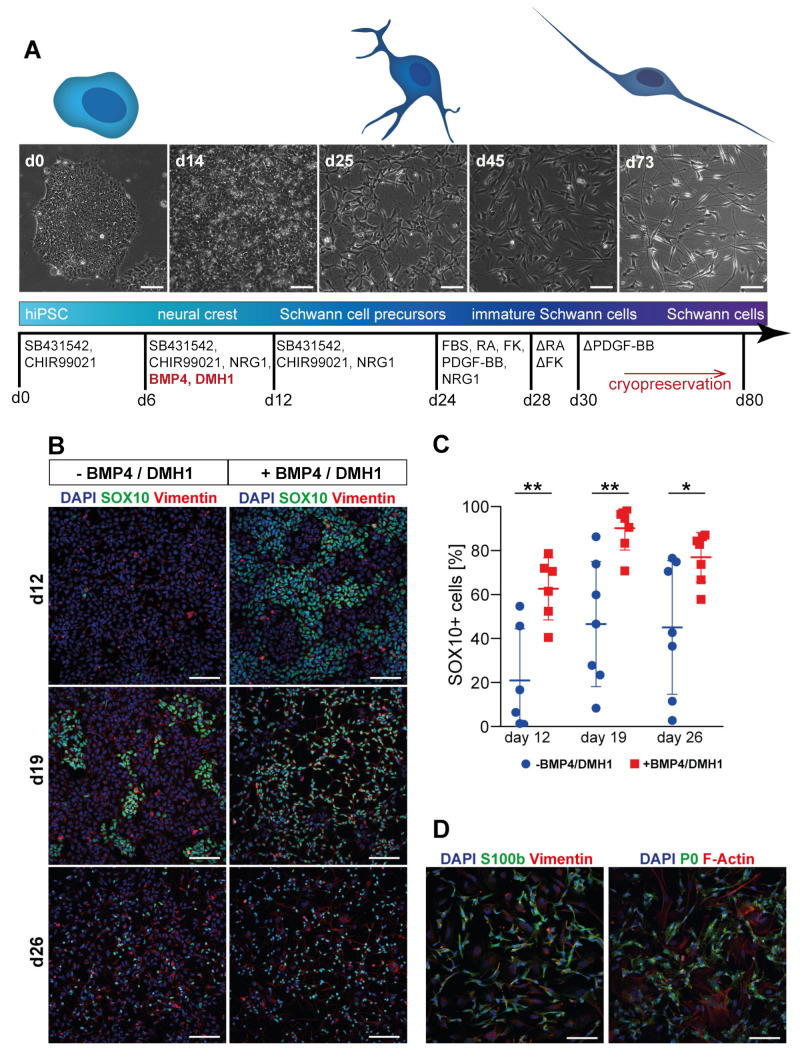
Robustness of Schwann cell differentiation is increased by defined activation of BMP signaling. (**A**) Schematic representation of protocol timeline and differentiation factors used. Micrographs show representative brightfield images of cell populations at differentiation times as indicated on the top left of each image. SC were obtained after 30–40 days of differentiation but could be cultured and further matured until at least day 100; cryopreservation could be performed at every passage. Scale bars: 100 µm. (**B**–**D**) hiPSC were differentiated according to the protocol shown in A. Control of BMP signaling was either applied (+ BMP4/DMH1) or left out (- BMP4/DMH1). Samples were fixed at time points as indicated and then immunostained for markers as displayed on top of panels. Nuclei were labeled with DAPI. (**B**) Representative micrographs showing fluorescence signals for SOX10 and Vimentin under conditions as indicated. Color code: see top of panel. Scale bars: 100 µm. (**C**) Quantification of SOX10+ cells under conditions and at time points as indicated. Graph depicts mean ± SD (n = 6 experiments for day 12; n = 7 experiments for day 19 and day 26). * *p* < 0.05, ** *p* < 0.01. (**D**) Representative micrographs of cells differentiated in the presence of BMP4 and DMH1 showing fluorescence signals for SC markers, S100b or myelin protein zero (P0), as well as Vimentin or F-actin at day 32. Color code: see top of panel. Scale bars: 100 µm.

**Figure 4 cells-10-03292-f004:**
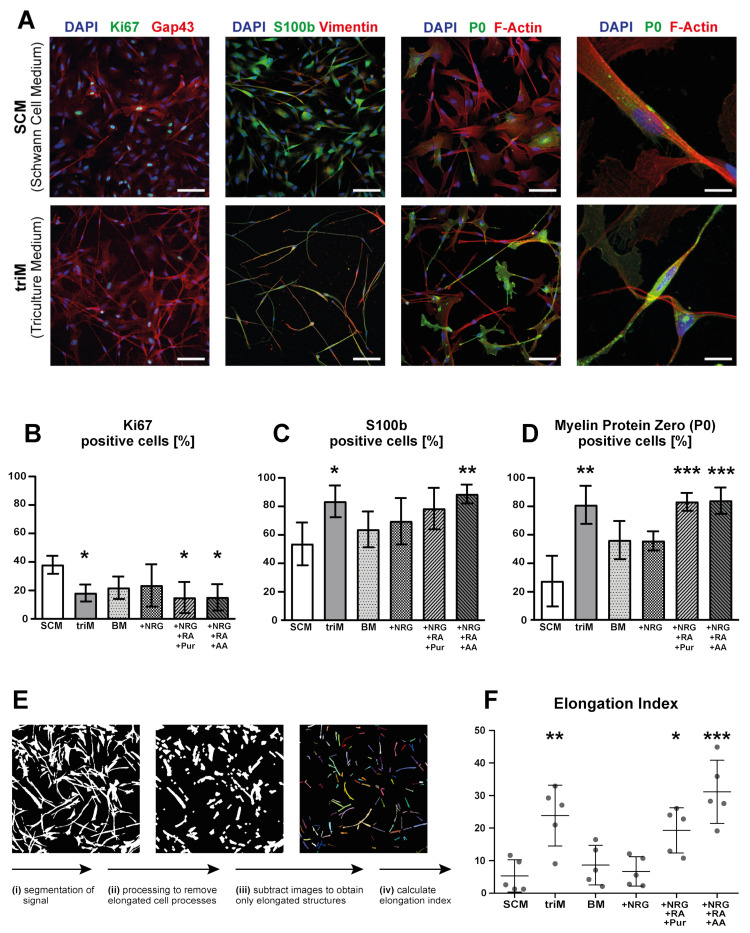
Triculture medium promotes differentiation of hiPSC-derived Schwann cells. SC differentiated from hiPSC using the SC differentiation protocol, including BMP4 tuning, were cultured for three days in either standard SC medium (SCM), fully supplemented triM, triculture basal medium (BM), or BM with supplementation of different factors (indicated by “+”). Then, cells were fixed and stained for different marker proteins and imaged using fluorescence microscopy. (**A**) Representative micrographs of cells stained for Ki67, Gap43, S100b, Vimentin, P0, and F-actin (color codes indicated on top of each panel); examples shown for SCM and triM conditions. Scale bars: 100 µm; zoom images in fourth column: 20 µm. (**B**–**D**) Quantification of cells positive for Ki67 (**B**), S100b (**C**), and P0 (**D**) for all conditions. Mean ± SD (n = 5 for Ki67 and S100b, n = 3 for P0). (**E**) Illustration of image processing workflow to calculate the elongation index. (**F**) Comparison of elongation index values obtained for all conditions. All values presented as mean ± SD (n = 5). * *p* < 0.05, ** *p* < 0.01, *** *p* <0.001.

**Figure 5 cells-10-03292-f005:**
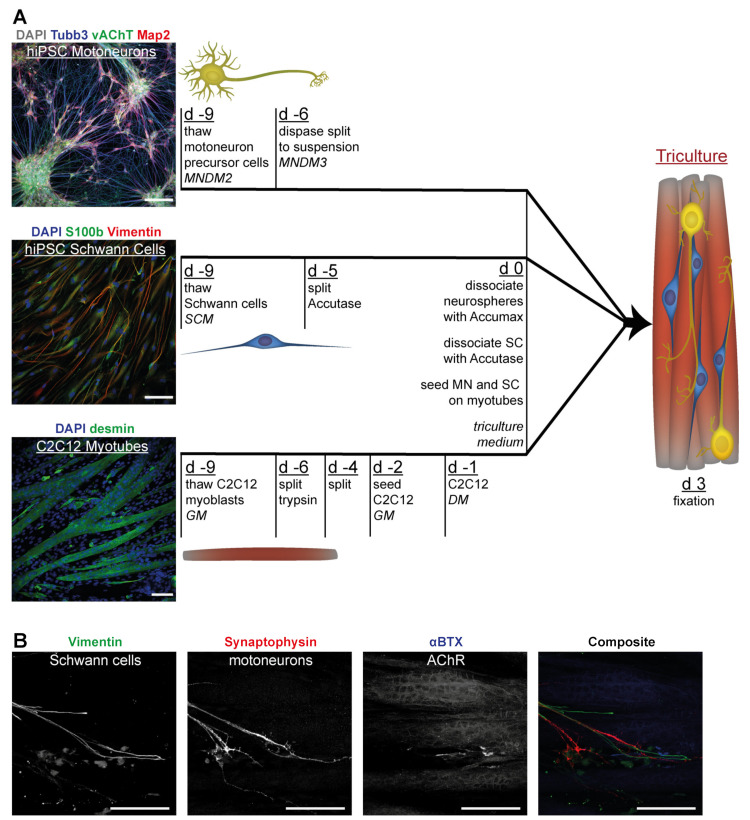
Tricultures can be set up from cell stocks within nine days. (**A**) Schematic timeline to set up tricultures, work steps, and different culture media shown. MNDM: MN differentiation medium; SCM: SC medium; GM: myoblast growth medium; DM: myotube differentiation medium. All medium compositions detailed in Methods section. Micrograph inserts depict fluorescence signals of representative individual cultures, respective stainings indicated on top of each panel. Scale bars: 100 µm. (**B**) Representative confocal image depicting SC (green in composite, Vimentin), MN (red in composite, Synaptophysin), and AChR clusters on C2C12 myotubes (blue in composite, αBTX staining). Scale bars: 50 µm.

**Figure 6 cells-10-03292-f006:**
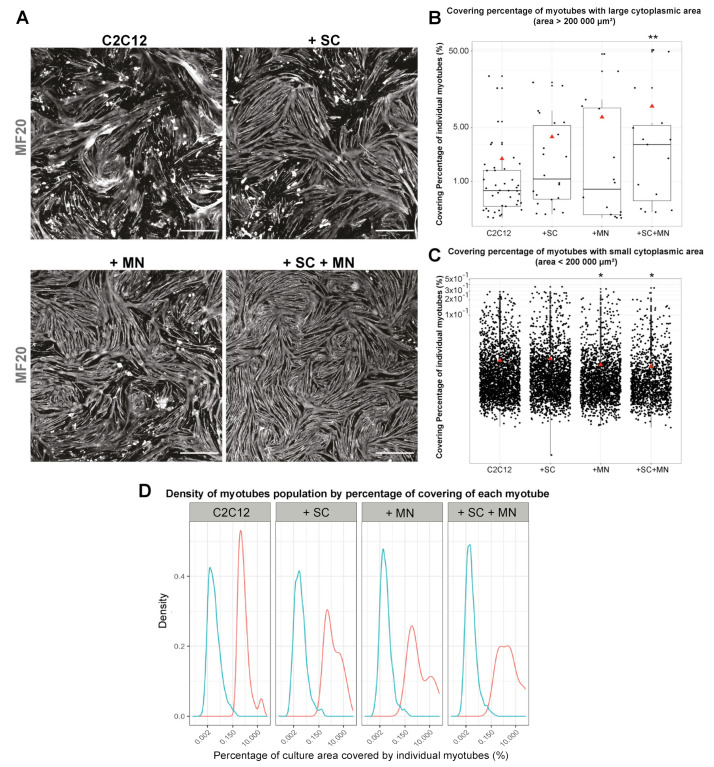
Cocultures promote formation of myotubes with increased cytoplasmic area. C2C12 cells were cultured on cover slips for three days in the absence (C2C12) or presence of either SC (+ SC), MN (+ MN), or both (+ SC + MN). Then, cells were fixed and immunostained for the muscle cell marker myosin heavy chain (MF20). (**A**) Representative microscopic overviews of MF20 fluorescence signals under culture conditions as indicated. Scale bars: 1000 µm. (**B**,**C**) Quantification of covering area of myotubes classified into two populations: large-type myotubes (>200,000 µm^2^) (**B**) and small-type myotubes (<200,000 µm^2^) (**C**). Red triangles on boxplots indicate mean values for given conditions; y-axes logarithmic. * *p* < 0.05, ** *p* < 0.01. (**D**) Density curves of analyzed myotubes expressed according to their respective percentages of covering area of the total cellular culture area. Blue and pink lines: density curves for myotubes with low and high cytoplasmic area, respectively; x-axes logarithmic.

**Figure 7 cells-10-03292-f007:**
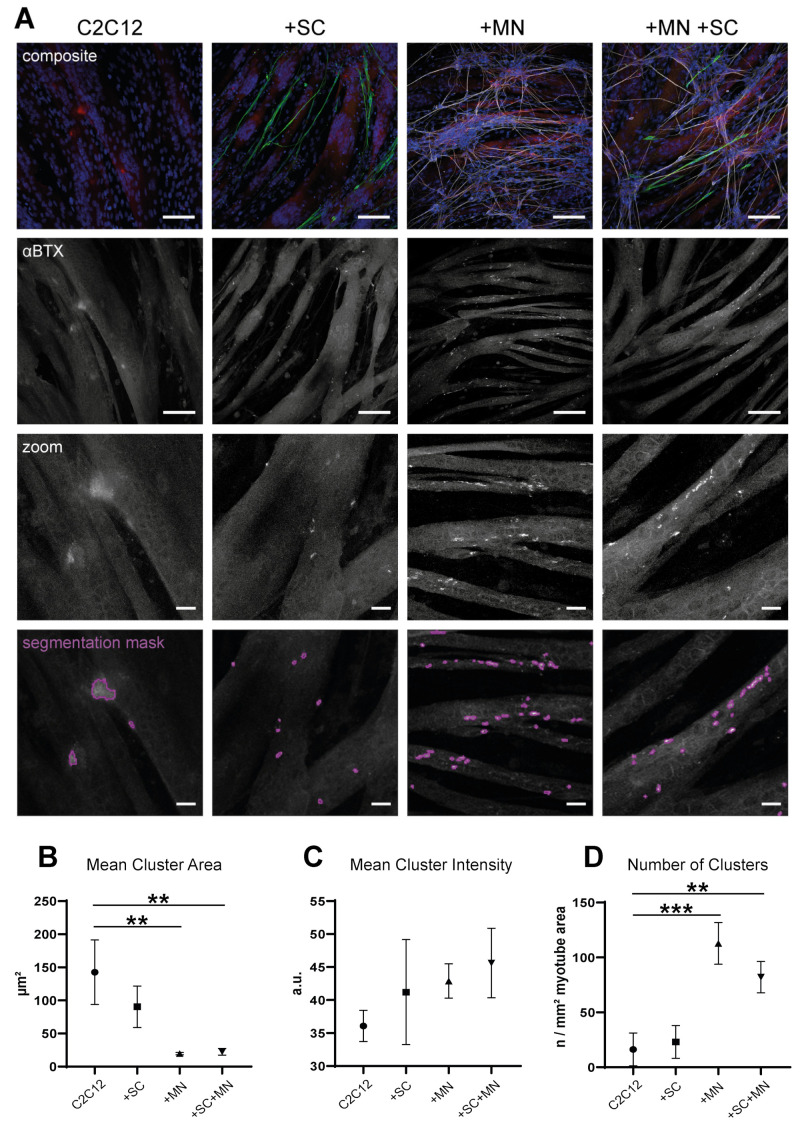
Cocultures with hiPSC motoneurons and Schwann cells influence formation of AChR clusters on C2C12 myotubes. C2C12 cells were cultured on cover slips for three days in the absence (C2C12) or presence of SC (+ SC), MN (+ MN), or both (+ SC + MN). Then, cells were fixed and stained for marker proteins, including AChR clusters. Upon confocal fluorescence microscopy, AChR clusters and their mean area, intensity, and number in the different coculture conditions were determined. (**A**) Representative confocal fluorescence images under culture conditions as indicated. Composite images showing fluorescence signals as follows. First row: DAPI (nuclei, blue), S100b (SC, green), βIII-tubulin (MN, gray), and αBTX (AChR, red); second row: αBTX staining only, in grays. Scale bars: 100 µm. Third and fourth rows show zoom images of representative AChR clusters and corresponding segmentation masks (purple outlines) as used for analysis. Scale bars: 20 µm. (**B**–**D**) Quantitative analysis of area of segmented AChR clusters (**B**), fluorescence intensity per segmented cluster (**C**), and number of segmented clusters normalized to myotube area (**D**) as a function of culture condition. All data, mean ± SD (n = 3). ** *p* < 0.01, *** *p* < 0.001.

## Data Availability

All experimental data will be available upon request.

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
