# Peer review of "hiPSC-Derived Schwann Cells Influence Myogenic Differentiation in Neuromuscular Cocultures"

_cells, 2021, doi:10.3390/cells10123292_

Round 1

Reviewer 1 Report

Manuscript ID cells-1453061, hiPSC-derived Schwann cells influence myogenic differentiation in neuromuscular cocultures by Hömer et al.

This manuscript reports an improved protocol to differentiate human iPSCs to Schwann cells in high yield. These hiPSC-derived Schwann cells enhanced myotube growth, and acetylcholine receptor accumulation on myoblast cell line C2C12 cells when the Schwann cells were cocultured with human iPSC-derived motor neurons and C2C12 cells. The manuscript examines the effect of Schwann cells in in vitro model of motor neuron and muscle cell coculture system. Still, it reports a limited amount of data about in vitro neuromuscular junctions. The manuscript is well written with details on the Schwann cell differentiation protocol, in vitro phenotype, and the effect of Schwann cells on muscle cell maturation. This manuscript is an interesting methodological manuscript and needs minor clarification for publication.

Line 399 “NMJ tricultures can be established from frozen cells in nine days”

  • The coculture preparation can be started in nine days, but in vitro NMJs do not seem to form in nine days. Therefore, this title is misleading and should be revised.
  • There is the same issue for lines 554 to 556.
  • In their title and abstract, the authors have been accurate and careful not to call their coculture system an in vitro NMJ model. This is a good approach with limited data showing the formation of NMJs in their culture system. Therefore, the section title “NMJ tricultures” should be revised not to include NMJ because the data in Figure 5B and supplementary figures only show weak colocalization of the three types of cells and do not test any further to confirm the establishment of NMJs using different methods.

Lines 402-403, “As for that, the best results were obtained when C2C12 were switched to triM after a short induction phase in C2C12 differentiation medium (data not shown).”

  • Explain how the results were evaluated and how the best condition decision was based on.

Lines 438-461, “Cocultures promote formation of myotubes with increased cytoplasmic area.”

  • Was there any difference in the cell density? Images in Figure 6A seem to show the morphological differences that the authors report and also the differences in cell number or density.

Author Response

Dear referee,

we are grateful for the constructive expert review. All comments were addressed. Please find a point-to-point response in the attached file.

Reviewer 2 Report

General comment.

The manuscript is written clearly, methods are detailed and the procedure to obtain the triculture is accurately described step by step. Cellular identity and maturation steps have been validated by imaging using specific markers and by morphological observations. A complete state of the art of the advantages/limitations of the already published in vitro NMJ models, and the improvements provided by the model presented here, are discussed as well.

Minor points:

- Please specify also in the Materials and Methods section the function of some reagents added to the culture medium (i.e. CHIR99021, SB431542, DMH1…)

- Could authors better explain the concept of BMP signaling tuning for neural crest specification? Although authors have discussed about it in paragraph 3.2, it is still not clear to me the way they found the right concentrations and timing to modulate BMP signaling (did they use the same dose/timing of Hackland’s work?).

- One major issue of in vitro NMJ models is the functionality of the culture system. What experiments could be planned to assess that the in vitro NMJ is indeed mature and functional? Are hiPSC- derived Schwann cells in the triculture able to trigger a plastic response (i.e. phagocytosis) upon NMJ injury as it occurs in vivo? Is a long-term nerve blockade able to induce AchR scattering? What are the exploitable read-outs to assess such properties of the in vitro NMJ model? Given the relevance of the issue, and also considering the future applications of the model, this deserves to be at least discussed.

Author Response

Dear referee,

thanks a lot for the positive and helpful comments. We have addressed all concerns. The point-to-point response can be found in the attached file.
